

# Observation of peri-implant bone loss rates in patients visiting dentist—A retrospective evaluation of patients of a faculty hospital for one year

Ilkim Karadag[1], Halis Kurnaz[1], Mehmet Murat Akkaya[2], İrem Karadag[3] and Zeynep Ilayda Konukçu Kurnaz[4]

[1] Periodontology/Faculty of Dentistry, Ankara University, Ankara, Turkey
[2] Periodontology/Faculty of Dentistry, Lokman Hekim University, Ankara, Turkey
[3] Prosthetics, Osmanlı Oral and Dental Health Center, Ankara, Turkey
[4] Prosthetics/Faculty of Dentistry, Ankara University, Ankara, Turkey

## ABSTRACT

**Background:** Many studies have been published on the relationship between different parameters with marginal bone loss around implants. The results of these studies vary, but some variables such as smoking or certain systemic conditions are more likely to predispose or exacerbate the resorption around implants. The aim of this study was to determine the rates of implants with radiographically detected marginal bone loss and to determine whether there is a relationship between the severity of destruction and certain risk factors like location of implant, restoration type, systemic condition, age, sex or smoking habits.

**Materials and Methods:** Panoramic radiographs obtained for 1 year were examined. Patients with bone loss around the implant were classified according to the region of implant placement, type of restoration on implants, systemic diseases, and smoking habits. The rate of bone loss around the implants was recorded as the resorption score. Double and multiple comparison tests were applied to observe whether the resorption scores were related to the variables.

**Results:** Of 17,352 patients, 1,465 had at least one implant, and 1,116 of these had no bone loss. A total of 181 patients (863 implants) included in the study, there was a weak correlation between age ($p = 0.017$) and resorption rate. Implants supporting bridge restorations had higher resorption scores. Gender, age, and systemic conditions alone are not effective in increasing peri-implant bone loss ($p < 0.05$); therefore, placing implants in the mandible ($p = 0.020$) or using implants to support bridge restorations($p = 0.027$) may make implants more vulnerable to resorption.

## INTRODUCTION

Implant-supported dental prostheses are constantly preferred by patients because they allow conservative prosthetic applications and provide rehabilitation without any

Corresponding author
Ilkim Karadag,
karadagilkim@gmail.com

intervention to the existing healthy teeth. The health status provided by implant-supported restorations is longlasting.

Although implant-supported dental restorations offer very successful results in short and long-term follow-ups, they are not exempt from complications. These are associated with inappropriate treatment planning, incorrect surgical technique or prosthetic application, material failures, and poor maintenance. Despite the high implant survival and success rates, it has long been recognized that osseointegrated implants can suffer from biological complications called peri-implant diseases (*Derks & Tomasi, 2015*). Peri-implant disease is a troubling and serious problem in dentistry, both therapeutically and epidemiologically. With the spread of implantology practice and the increase in the number of implants placed every year, the frequency of peri-implant disease is greatly increasing (*Lee et al., 2017*).

Although it is not a globally used classification for peri-implant diseases, peri-implant mucositis and peri-implantitis, are equivalents to gingivitis and periodontitis, respectively (*Renvert et al., 2018*). Peri-implantitis is an irreversible disease that causes resorption of the bone tissue surrounding the implant and the formation of pathological pockets around the implant. In the consensus report of the 2017 World Workshop, peri-implantitis is described as "a plaque- associated pathological condition occurring in tissues around dental implants, characterized by inflammation in the peri-implant mucosa and subsequent progressive loss of supporting bone" (*Berglundh et al., 2018*). As in the gingivitis-periodontitis relationship, peri-implantitis also starts with peri-implant mucositis, leading to resorptions in the bone tissue surrounding the implant unless the cause is eliminated (*Berglundh et al., 2018*).

Peri-implant bone loss is the most important criterion that distinguishes peri-implantitis from peri-implant mucositis. If left untreated, it can lead to pain, loss of function, and loss of the implant. The other points to be considered are the bone loss resulting from implant failure before or after loading, bone loss resulting from surgical milling of the bone during implant removal or the widening of the existing implant socket in the bone crest as a result inflamations occurring for various reasons. It is even possible that the cortical bone can completely disappear starting from the fenestration in the outermost cortical layer of the crest or the defect can turn into a defect with three bone walls, which may lead to the use of additional biomaterials by performing additional surgical procedures for bone filling of these defects or implant replacement. These situations cause serious expenditures of time, money and resources. In order to prevent these situations, correct planning, appropriate surgical method and prosthetic restoration produced with high precision are the main research subjects of implantology. Another recommended point is to call patients for routine check-ups and monitor their general oral health as well as the condition of the soft tissue and bone tissue surrounding the implants in use. *Albrektsson et al. (1986)* recommended that a successful implant should be free of mobility, peri-implant radiolucency, radiographically detectable bone loss greater than 0.2 mm per year after the first year of loading, and no persistent pain, discomfort, or infection. This is the most commonly used method for evaluation of dental implants.

Although the primary etiological cause is biofilm, many risk factors may contribute to the development of peri-implantitis. Among these factors, the most clearly shown relationship with peri-implantitis in many studies and the workshop report presented as a result of these studies are the presence of diabetes mellitus (DM), smoking and history of periodontal disease (*Lee et al., 2017*; *Berglundh et al., 2018*; *Giok, Veettil & Menon, 2024*; *Berglundh et al., 2024*).

Peri-implant diseases are processes triggered by the accumulation of biofilm on the implant surface. Although the primary etiological cause is biofilm, many risk factors may contribute to the development of peri-implantitis. Among these factors, the most clearly shown relationship with peri-implantitis in many studies and the workshop report presented as a result of these studies are the presence of DM, smoking and history of periodontal disease (*Lee et al., 2017*; *Berglundh et al., 2018*; *Giok, Veettil & Menon, 2024*; *Berglundh et al., 2024*). As in periodontal diseases, the primary goal in treatment of peri-implant diseases is to control the plaque biofilm appropriately in order to stop the infection (*Arciola, Campoccia & Montanaro, 2018*). In peri-implant diseases, the macro and micro rough structure of the implant surface prevents the biofilm from being adequately removed from the surface (*Daubert & Weinstein, 2019*). A different treatment approach is adopted in the treatment of peri-implantitis compared to periodontitis due to the difficulties in both removing the biofilm from the implant surface at an adequate level during treatment and maintaining this with the patient's routine oral hygiene practices. Survival rates can be increased in implants suffering from peri-implantitis with some physical and chemical decontamination procedures performed on the implant surface (*Schwarz et al., 2013*). While plastic curettes, plastic-tipped ultrasonic scalers and titanium brushes can be used for physical cleaning, the use of air-abrasive systems or laser irradiation has been accepted as methods that provides beneficial results in terms of physical cleaning of the surface (*Schwarz, Becker & Renvert, 2015*; *Cosgarea et al., 2023*). In addition, chemical cleaning of the root surface with chemical agents such as citric acid (CA), chlorhexidine (CHX), hydrogen peroxide (HPO) or doxycycline gives positive results (*Gosau et al., 2010*). These methods can be applied alone on the implant surface, but the recommended method of use is to use them in combination with physical cleaning methods on the surface of the implant.

The importance of measuring bone levels around the dental implant in the diagnosis of peri-implant diseases on radiographs is well understood. Panoramic radiographs obtained routinely from patients who apply to the dentist with any dental problem are also suitable for this purpose (*Sadik et al., 2023*).

The aim of this article is to determine the rate of radiographic evidence of bone loss in patients with dental implants that have been in use as a fixed restoration support for at least 3 years among patients admitted to our hospital for a year, and to reveal data on the relationship between the prevalence of bone loss around the implant and some risk factors for peri-implantitis.

## MATERIALS AND METHODS

### Patient selection

This study complies with the Declaration of Helsinki and was performed according to ethics committee approval (Ankara University Faculty of Dentistry Clinical Ethics Committee: 36290600/50). Written consent was not obtained from the participants because the design of our study was retrospective and patients were contacted *via* telephone. All patients who had panoramic radiographs in our hospital between January 01, 2021 and January 01, 2022 were evaluated with a preliminary examination. Panoramic radiographs were taken using the same device and same configuration (Promax 3D Max, Planmeca Oy, 00880 Helsinki, Finland. 54–84kV, 1–16 mA, 13.8 s). After being evaluated as described under heading "Patient Selection", implant patients with bone loss were contacted by registered phone numbers and asked to answer the following questions.

- How many years have you been using the prosthesis made on the implant?
- Do you have a systemic disease? If yes, which ones are they and are you under medical supervision for this condition?
- Have you undergone radiation therapy in the head and neck area before?
- Do you smoke? If yes, on average, how many cigarettes do you smoke per day?

Age, gender, systemic disease information, average number of cigarettes consumed per day, total number of implants, and number of implants with bone loss were recorded for all patients. While the type of prosthetic restoration carried by each implant was recorded as crown or bridge, the area where the implant was located was divided into eight groups as maxillary incisive, maxillary canin, maxillary premolar, maxillary molar, mandibular incisive, mandibular canin, mandibular premolar, and mandibular molar.

### Inclusion criteria

- Patients over the age of 18.
- Patients who have been using an implant-supported fixed restoration for at least 3 years.
- Patients whose radiographs were obtained between January 01, 2021 and January 01, 2022.

### Exclusion criteria

- Patients using implant-supported removable dentures.
- Patients with implants placed in the last 3 years.
- Patients who cannot be contacted and who do not remember the implant installation date.
- Patients with radiographs in which the image quality does not allow evaluation.

### Evaluation of radiographs

Single investigator carried out all evaluations of panoramic radiographs in a quiet and dimly lit room using Sisopacs software (Sisoft-Ankara, Turkey). When evaluating marginal bone defects, whichever was deeper, mesial, or distal, was recorded as baseline (*Fransson et al., 2005*). This evaluation was performed according to the ratio of the amount of bone loss measured on the radiograph to the length of the implant measured on the radiograph.

Marginal bone loss classification was performed through using the following method recommended by *Froum & Rosen (2012)*.

Score 0: No visible marginal bone loss around the dental implant.

Score 1: Marginal bone loss less than <25% of the total length of implant measured on radiograph.

Score 2: Marginal bone loss at a level between 25% and 50% of the total length of implant measured on radiograph.

Score 3: Marginal bone loss more than 50% of the total length of implant measured on radiograph.

The bone loss score determined during the radiographic evaluation allowed us to numerically record the rate of bone loss around each implant which was recorded as the resorption score (RS) of that implant. To calculate the patient-based resorption rate the following calculation was performed: The sum of the RSs of all implants in the mouth was written in the numerator of the fraction. At this point, healthy implants without noticable resorption were involved into calculation as "0 resorption score". Three times the number of implants that the patient owned is written as the denominator, since the marginal bone loss score 3 is the highest possible value. Finally, the fractional value found was multiplied by 10 to obtain a numerical value ranging from 0 to 10, showing the total resorption score (TRS) for a patient.

$$TRS = \frac{Sum\ of\ the\ RSs\ of\ all\ implants}{Number\ of\ implants\ \times 3} \times 10.$$

## Statistical analysis

All obtained data were processed on a computer using a data analysis program (IBM, SPSS for Windows V22.0; Armonk, NY, USA). Before all comparisons, the normality of data distributions were tested using the Kolmogorov-Smirnov test. While the independent sample t- test was used for systemic disease and gender-based comparisons, comparisons based on whether the implant was located in the lower jaw/upper jaw and type of restoration (crown/bridge) were executed with the Mann–Whitney U test. The data distribution of the patient groups of cigarette consumption was not normal; therefore, they were evaluated with Kruskal–Wallis and Bonferroni correction was used to evaluate the significance of the difference for dual comparisons. The comparison of the groups separated according to the region of the implant was performed using one-way ANOVA, and Bonferroni correction was used for *post hoc* evaluations. Spearman's rank correlation test was used to determine the relationship between age and marginal bone loss ($p < 0.05$). Among all evaluated parameters, the combined factors of those shown to have an effect were re-evaluated with multivariate regression analysis.

## RESULTS

The data supporting the findings of this study are available in the Supplemental Material of this article. Panoramic radiography images of 17,352 patients were evaluated. Of these films, 1,465 (8.44%) had at least one implant (a total of 4,681 implants). While 1,116

**Table 1 Distribution of the implants according to specific location and the mean resorption scores of the groups.**

| Specific location of implant | N | Mean resorption score |
| --- | --- | --- |
| Maxiller incisor | 40 (6.36%) | 1.90 ± 0.12 |
| Maxiller canin | 41 (6.52%) | 1.85 ± 0.12 |
| Maxiller premolar | 109 (17.33%) | 1.85 ± 0.80 |
| Maxiller molar | 109 (17.33%) | 2.03 ± 0.83 |
| Mandibular incisor | 17 (2.70%) | 2.06 ± 0.22 |
| Mandibular canin | 16 (2.54%) | 2.06 ± 0.23 |
| Mandibular premolar | 79 (12.56%) | 2.13 ± 0.09 |
| Mandibular molar | 218 (34.66%) | 2.02 ± 0.05 |

(76.17%) patients (3,039 implants) did not have an implant with radiographically visible bone loss, 349 (23.83%) patients (1,642 implants) had at least one implant with radiographic bone loss. After telephone interviews, 168 patients (94 women, 66 men; 779 implants) were excluded for the following reasons:

- 72 patients had implants placed in the last 3 years.
- 16 patients did not accept participation in the study.
- 80 patients could not be reached from their registered phone numbers.

A total of 863 implants in 181 patients (98 women, 83 men) were included in the study. Although marginal bone loss was observed around 629 implants (72.88%) in these patients, no bone loss was detected in 234 implants (17.12%). It was learned that 79 of 629 implants with bone loss were removed because of poor prognosis (12.55%). The distribution of TRS scores were as following; "1" score was measured in 218 implants (34.66%), "2" score was measured in 199 implants (31.64%) and "3" score was measured in 212 implants (33.70%). The mean TRS scores were 5.30 ± 2.56 for females and 5.70 ± 2.65 for males. The subgroups of men and women were compared using the independent sample t-test, and there was no significant difference in the resorption score ($p = 0.593$).

The ages of 181 patients were ranged between 26 and 80 years (mean 55.54 ± 11.16). Spearman's rank correlation test results showed a significant but very weak correlation between age and bone loss rate around the implant ($r = -0.177$, $p = 0.017$). 52.62% of the implants included in the study were located in the mandible, and the mean TRS was 2.06 ± 0.45, whereas 47.38% were located in the maxilla (mean TRS = 1.91 ± 0.48), and it was observed that the implants located in the maxilla had a lower bone loss score than those located in the mandible ($U = 54,324.5$; $p = 0.020$). When the implants were classified according to the region, there was no significant difference between the eight groups in terms of resorption scores ($F = 1.076$, $p = 0.377$) (Table 1).

After classifying the implants according to whether they support a crown (103 implants, 16.38%) or bridge (526 implants, 83.62%) restoration, it was observed that the resorption scores of the implants supporting the bridge restorations (TRS = 2.02 ± 0.36) were higher than those of the implants used as support for crown restorations (TRS = 1.83 ± 0.75;

**Table 2 Distribution of the patients by daily cigarette consumption and mean total resorption scores.**

| Smoking habits | N | Mean resorption score |
|---|---|---|
| 0 (not a smoker) | 112 (39.58%) | 5.06 ± 2.51 |
| 1 (1–10 cigarettes per day) | 33 (11.66%) | 5.82 ± 2.82 |
| 2 (11–20 cigarettes per day) | 29 (10.25%) | 6.27 ± 2.41 |
| 3 (>20 cigarettes per day) | 109 (38.51%) | 2.03 ± 0.83 |

**Table 3 Pairwise comparisons of TRS according to smoking habits.**

| | Daily cigarette consumption groups | | | | | |
|---|---|---|---|---|---|---|
| | 0–1 | 0–2 | 0–3 | 1–2 | 1–3 | 2–3 |
| $p^{\beta}$ | 0.160 | 0.016 | 0.019 | 0.384 | 0.127 | 0.326 |
| Adj. $p^{\varphi}$ | 0.962 | 0.099 | 0.114 | 1 | 0.762 | 1 |

**Note:**
$\beta$: Independent sample Kruskal-Wallis test result ($x^2 = 10.639$). $\varphi$: Pairwise comparision of the TRS difference between the groups according to the amount of cigarettes consumed daily after Bonferroni correction ($p < 0.05$).

**Table 4 Results of multivariate regression analysis of relationship between resorption score and implant location according to jaw, restoration type and cigarette consumption.**

| Model | R | $R^2$ | Adj. $R^2$ | Sig. F change |
|---|---|---|---|---|
| 1 | 0.093[a] | 0.009 | 0.007 | 0.020 |
| 2 | 0.130[b] | 0.017 | 0.014 | 0.022 |
| 3 | 0.210[c] | 0.044 | 0.040 | 0.000 |

**Note:**
[a]: Predictors: maxillar or mandibular placement of implants (max/mand). [b]: Predictors: max/mand, restoration type (crown/bridge). [c]: Predictors: max/mand, crown/bridge, cigarette consumption.

U = 30,599.0; $p = 0.027$). Table 2 shows the data of the groups classified according to the daily smoking habits of the patients. The data distribution was not normal (Kolmogorov-Smirnov $p = 0.014$) and independent sample Kruskal-Wallis test were carried out. Higher resorption scores were obtained in the groups with 10 cigarettes smoked daily, but this difference was not statistically significant (Table 3). Taking these results into consideration, the results of the regression analysis, in which we conducted a more detailed analysis of the relationship between the resorption score and the jaw where the implant is placed, the type of restoration it supports and smoking, are shown in Table 4.

Patients were grouped in terms of being diagnosed with diabetes mellitus, cardiovascular disorders, and thyroid disease (Table 5). In addition, it was learned that five patients (five implants) had rheumatoid arthritis and one patient (one implant) had a history of cancer; however, these patients were not subject to a separate classification because these quantities are not viable for a healthy statistical analysis. None of the patients included in the study had previously received radiotherapy or postmenopausal hormone replacement therapy or had a history of osteoporosis.

**Table 5 The significance of the difference found after comparing the resorption scores of the sample groups with systemic disorders using the independent sample t-test.**

| Systemic condition | | N | TRS | p |
|---|---|---|---|---|
| Diabetes mellitus | Yes | 23 (12.71%) | 4.85 ± 2.54 | 0.692 |
| | No | 158 (87.29%) | 5.58 ± 2.60 | |
| CVD | Yes | 42 (23.20%) | 5.21 ± 2.57 | 0.928 |
| | No | 139 (76.80%) | 5.57 ± 2.61 | |
| Thyroid disease | Yes | 11 (6.08%) | 4.09 ± 2.52 | 0.806 |
| | No | 170 (93.92%) | 5.58 ± 2.59 | |

Of the 42 patients in the cardiovascular diseases (CVD) group, 29 had hypertension only and 13 had coronary artery disease. No relationship was found between diabetes, CVD and thyroid disease, and bone loss (Diabetes $p = 0.692$, CVD $p = 0.928$, thyroid diseases $p = 0.806$).

## DISCUSSION

The most common biological complication affecting dental implants is peri-implant disease. It should not be forgotten that when peri-implant diseases develop, loss of function, time, labor, and financial losses from non-surgical or surgical treatments are experienced. Our results showed that the incidence of peri-implant bone loss increases with age, and implants located in the mandible or used as bridge restoration support are more prone to peri-implantitis. Other factors found to be associated with bone loss were smoking and the presence of DM. Many research results have been published on marginal bone loss, incidence of peri-implantitis, and short/long term implant survival around dental implants (*Atieh et al., 2013*). Although an important part of the studies on this subject are prospective or cross-sectional, prevalence studies on the subject have also been published, albeit with a smaller proportion (*Berglundh, Persson & Klinge, 2002*). Renvert reported that an average of 54.7% of the implant patients they evaluated with a follow-up period of 21–26 years were diagnosed with peri-implant mucositis and 22.1% with peri-implantitis, and noted that the diagnosis and occurrence of these two conditions was high (*Renvert, Lindahl & Persson, 2018*). Another more recent study evaluated 490 implants in 147 patients (*Pimentel et al., 2018*). In this study, in which a sample size similar to our study was evaluated, it was reported that peri-implant mucositis was observed in 85.3% of the implants evaluated and peri-implantitis was observed in 9.2%.

Implant survival studies are designed to monitor/evaluate changes up to a specified time after implantation in relatively smaller sample groups because of the difficulty in achieving patient standardization, and the success of implant treatments is largely dependent on the success of the management of local factors (*Carr, 2012*). Although the results are clearer and more precise, the evaluation of a controlled group with certain characteristics can be seen as a weakness of such studies. We designed our study by prioritizing answering the question of "how often peri-implant marginal bone loss can be observed in patients who visit the dentist for any reason". Developing this question as "What is the incidence of

peri-implantitis or peri-implant mucositis in addition to marginal bone loss" and designing a cross-sectional study would allow the comparison of clinical findings with radiographic evaluation and testing the reliability of radiographic findings in peri-implant areas as well as marginal bone loss. It will also pave the way for the direct evaluation of the disease instead of disease symptoms.

Since we focused on determining the ratio of patients with peri-implant marginal bone loss among patients with implants during the design phase of the study, we did not collect data to properly evaluate implants that received peri-implantitis treatment. It should not be forgotten that even implants with marginal bone loss or even those whose body is exposed to the oral environment can continue to function without causing discomfort to the patient with the treatment applications to be carried out (*Berglundh et al., 2018*). It has been proven that biofilm accumulation can be prevented on the implant surface thanks to the physical and chemical decontamination methods carried out for this purpose. However, the clinician must work with high sensitivity while applying these methods, because incorrect applications that can be made with the aim of cleaning the implant surface are likely to cause the situation to worsen rather than improve it. Studies have shown that using curettes made of materials that will not cause unwanted scratches on the metal surface, such as plastic or its derivatives, instead of metal hand tools for instrumentation on the implant surface has a more positive effect on the implant survival rate in the long term. Similarly, it has been shown that the use of high concentrations of CA used for chemical decontamination damages the titanium oxide layer on the implant surface under the effect of low pH, especially in the method of cleaning the surface with rubbing (*Wheelis et al., 2016*). While the titanium oxide layer covering the implant surface is the most important element that protects the implant surface against corrosion, if it is damaged, corrosion and separation from the surface, called "pitt attack", are inevitable (*Suito et al., 2013*; *Valderrama et al., 2014*). As a result, a rough structure can be formed on the surface that will facilitate biofilm accumulation. It has been indicated that the 40% concentration of CA, which is the most frequently used chemical agent for detoxification purposes, removes bacterial biofilm from titanium implants at a high rate, but since it also damages the surface, it has been understood that the situation may evolve into a situation far from what is desired in terms of preventing biofilm accumulation in the long term. In addition, the negative structural change of titanium surface, the failure of prosthetically loaded implants to withstand possible occlusal forces and the formation of fractures or cracks in the implant neck and body are among the most serious complications that can be seen. Although there are many studies and meta-analyses evaluating the effectiveness of peri-implant treatments, conducting epidemiological studies to investigate the rates of improvement or additional bone loss in treated peri-implantitis cases and evaluate the effects of etiological factors for peri-implant diseases would provide deeper information on the treatment of peri-implant diseases, which is a constantly evolving concept in the light of constantly renewed information, and the maintenance of the achieved health status.

The results of animal experiments and cross-sectional human studies have shown that the microorganisms associated with periodontitis and peri-implantitis are similarly gram negative anaerobic bacteria. Furthermore, it is known that biopsies taken from

periodontitis and peri-implantitis lesions show similar characteristics. Basically, similar biomarkers are regulated in both diseases (*Schwarz et al., 2018*). Although a similarity is seen at this level, the rate of disease progression and the severity of the inflammatory response may be different. The probable reason why tissue destruction is more pronounced in peri-implantitis clinically and radiographically is that implants do not have collagen fiber bonds like natural teeth. It has also been claimed that the collagen capsule protecting the supracrestal fibers helps to keep the lesions more limited by entering between the lesion and the alveolar bone, and that the absence of this structure in the implant perimeter may also explain the rapidly progressing nature of peri-implant lesions.

Although standardization of the patients is a factor that increases the reliability of the studies, it is a more appropriate approach to limit the compared parameters instead of standardizing them to ensure standardization in studies involving many patients and in which the prevalence is investigated, such as in our study (*Wears, 2015*). With this in mind, in our study, instead of qualifying the marginal bone loss around the implant with millimetric measurements or the number of threads adjacent to the loss, we benefited from an index with less numerical precision but also reliable at different magnifications. Different methods have been developed to assess and score peri-implantitis, or specifically bone loss. Similar to our study, *Wada et al. (2019)* in order to measure the prevalence of peri-implant diseases and risk factors in implants that have been functioning for at least 3 years, compared the distance between the most coronal point of the bone adjacent to the implant and the apical end of the implant on intra-oral radiographs and the actual length of the implant. Their results show that the prevalence for peri-implant mucositis is 27.4% and the prevalence for peri-implantitis is 9.2%.

The marginal bone loss index used in this study was a parameter evaluated in the peri-implantitis classification introduced by *Froum & Rosen (2012)*. The original version of the classification can be defined as a combination of evidence of bleeding and/or suppuration on probing, probing depth, and the degree of radiographic bone loss around the implant. It is used to categorize the severity of peri-implantitis into early, moderate, and advanced phases.

In addition, we think that there are some advantages to using the classification proposed by *Froum & Rosen (2012)* instead of millimetric measurements for the assessment of bone loss around the implant. Bone loss starts with prosthetic loading, it is a condition that can progress over time without peri-implantitis, and some resorption can be considered as "physiological" (*Albrektsson et al., 1986*). A relatively small bone loss measured in a short implant will have a more negative prognosis than the same amount of bone loss in a longer implant. This means that the millimetric value recorded for implants that have suffered bone loss that does not even require treatment may lead to "selective surveillance bias".

According to the results reported by *Doornewaard et al. (2018)* biological parameters such as probing depth and bleeding on probing may not always be correlated with peri-implant bone loss. Unlike the classification referenced in our study, we only recorded the amount of marginal bone loss around the implants according to the index value and reached a numerical value that allowed us to conduct statistical analysis.

Because our study was a retrospective study, it was not possible to determine the parameters that would increase standardization, such as the single brand of the evaluated implants or the inclusion of implants of a certain size, as inclusion criteria. We did not prefer to standardize implants because it would cause a serious decrease in the number of patients and implants taken into account, and it would negatively affect the reliability of the statistical results to be obtained.

In terms of the results of bone loss, it is possible to compare the radiological methods used in the diagnosis of peri-implant diseases and to come across different results. *Kühl et al. (2016)* compared two-and three-dimensional radiographic images and reported that intraoral radiographs are more useful than cone-beam computed tomography (CBCT). *Ritter et al. (2014)* reported that there was no difference between two- and three-dimensional imaging studies in terms of mesial and distal bone level measurements, but reported that CBCT was more effective in terms of evaluations of chemicals found in the buccal and oral faces. While studies have shown that panoramic radiographs can give erroneous results in the measurement of peri-implant bone loss, there are also articles stating that panoramic radiographs are an adequate tool to evaluate peri-implant bone loss (*Sadik et al., 2023*; *Tercanli Alkis & Turker, 2019*).

Many clinical studies have evaluated the relationship between implant success and the time elapsed between implant placement and prosthetic loading (*Alkan et al., 2018*). It has long been recognized that peri-implant marginal bone loss tends to increase over the years (*Derks & Tomasi, 2015*; *Vázquez Álvarez et al., 2015*). This occurs because of physiological remodeling of the marginal bone following implant placement and prosthetic loading. In our study, we did not evaluate the effect of the time that the implants remained in the mouth on marginal bone loss. The reason for this is that the patients included in the study also had implants made in different regions of the mouth on different dates, and not all patients could remember the exact date of the treatment procedures.

We can interpret the reason for the very weak correlation we found between the prevalence of peri-implant diseases and age as the increase in people receiving implant treatment to compensate for physiological tooth loss as age increases, the increase in the average number of implants per person, and the increase in the incidence of systemic diseases that are likely to accompany older ages. In a study evaluating risk factors for peri-implant diseases, it was shown that age was not a directly affecting factor with the data collected from 40 patients followed (*Benedek et al., 2024*). Unlike our study, the authors grouped the patients according to their ages instead of using correlation tests to evaluate the age/peri-implantitis relationship. *Hussain et al. (2024)* compared implants of two different brands in 14 patients in terms of risk factors for peri-implant diseases and could not detect a significant relationship between age and peri-implantitis. A review also revealed that there is moderate to high evidence of a relationship between patient age and peri-implantitis (OR = 1.0, 95%, CI [0.87–1.16]) (*Dreyer et al., 2018*). The reasons for the differences in the results can be considered as methodological differences due to the different primary objectives of the studies and relatively small sample groups or meta-meta analysis data being subjected.

It is known that there are a number of risk factors for peri-implantitis. A recent meta-analysis study on this subject evaluated 41 unique risk factors and showed that 24 factors were associated with peri-implantitis. Again, the results of this analysis indicated that the highest associated risk factors for peri-implantitis were the presence of periodontitis and smoking (*Giok, Veettil & Menon, 2024*). Due to the retrospective nature of our study, we did not have sufficient data to evaluate whether there was a relationship between the presence of current periodontitis or past history of periodontitis and the occurrence of peri-implant diseases. Considering that history or existence of periodontitis is the most important risk factor, it would be more accurate to evaluate the issue with prospective studies.

Some studies have shown that smoking is the most prominent risk factor for peri-implant disease (*Cavalli et al., 2015*). *Clementini et al. (2014)* stated that smoking increases peri-implant bone destruction by 0.16 mm per year. *Gupta et al. (2018)* showed that smoking may be an important factor in implant failure. However, the relationship between smoking habits and peri-implantitis remains controversial, as some patient-based studies did not reveal significant differences in the risk of peri-implantitis between smokers and non-smokers (*Sgolastra et al., 2015*). According to the results of our study, the rate of resorption tends to increase as daily cigarette consumption increases. In particular, the difference in TRS between non-smoker patients and those who smoked 11–20 cigarettes a day was the highest. In addition, because of the retrospective nature of our study, it was not possible to distribute an equal number of patients to the groups determined according to the amount of cigarettes consumed per day. To evaluate the effect of cigarette consumption and alcohol or other substance addictions on peri-implant bone loss more accurately, observational studies should be conducted by taking these criteria into account when selecting the patient population.

Among all risk factors, history of periodontal disease, smoking and the presence of DM were generally the factors most clearly associated with peri-implantitis (*Renvert et al., 2014*). During the analysis of the data we collected in the sample group, instead of evaluating all systemic diseases, we preferred to evaluate DM, which has been proven to be associated with peri-implantitis, and hypertension and thyroid disease, which are highly prevalent in the society. Total of five implants included in the study were in patients with rheumatoid arthritis and one implant was in patients with a history of cancer. Since these figures would not allow a healthy analysis, a separate subgroup was created and not evaluated. Similar to our study, our sample group had thyroid disease. In a study that evaluated the relationship between peri-implantitis and all accepted risk factors and predisposing factors with data collected from 916 osseointegrated implants in 183 patients, peri-implantitis was detected in 16.4% of the patients (7.3% of the patients) (*Dalago et al., 2017*). The incidence of peri-implantitis was found to be significantly higher in people with heart diseases, stomach and thyroid diseases.

Another risk factor known to be associated with peri-implantitis is the presence of DM. Only 23 of the 181 patients included in our study had DM. Although the average RS value calculated in the diabetic people we included in the statistical analysis was lower than in the non-diabetic group, no significant difference was detected between the two groups.

However, due to unequal sample distribution, it is far from being a definitive test that evaluates the relationship between peri-implantitis and the presence of DM. Also, although all diabetic patients stated that their blood sugar levels were under control, we did not have data on how much blood sugar control was achieved at the time of surgery and in the postoperative period. While some articles published to date have not shown any significant relationship between the presence of DM and implant success, the results of other studies have revealed that the presence of DM poses a high risk for peri-implant marginal bone loss (*Zupnik et al., 2011*). *Khandelwal et al. (2013)* reported that a 98% success rate was achieved in individuals with poor diabetic control, even if some complications were experienced. The findings of another recent study also showed that there is a bidirectional relationship between DM and peri-implant diseases, meaning that controlling one of these conditions contributes significantly to controlling the other condition (*Enteghad et al., 2024*).

Similar to diabetes, the results of studies investigating the relationship between the presence of CVD and bone loss around the implant also point to different aspects. While there are studies showing that there is no correlation between the presence of CVD and peri-implant bone loss, there are also publications showing that peri-implantitis is associated with the presence of CVD (*Renvert et al., 2014*). The mechanism by which cardiovascular disorders compromise the osseointegration and healing process can be explained by a poor blood flow that may restrict the delivery of oxygen and nutrients to the tissues (*Krennmair et al., 2016*; *Hwang & Wang, 2007*). Our study included 42 patients with CVD, and no significant relationship was found between CVD and peri-implant bone loss. Choosing study designs in which the effects of different levels of blood pressure irregularity and the effect of using different hypertension drugs are being evaluated could reinforce the ability to reveal more accurate findings.

It was not possible to divide all implants into subgroups according to the period of time they functioned in the mouth, because the rate of patients' implants being placed at different times was very high and not all patients could remember the exact date their implants were placed during the telephone conversation where the information was obtained. In their study, *Pimentel et al. (2018)* showed that the risk of peri-implantitis increases two times in implants that have been functional for 5 years or more. In a similar study, peri-implantitis was detected at a rate of 38.4% in implants that remained functional for 10 years or more, while this rate was calculated as 17.6% in implants that remained functional for less than 10 years (*Marrone et al., 2013*). It is clear that with a prospective study design in which questions can be asked face to face to patients rather than meeting on the phone, clinical findings will also be taken into account and it will be easier to detect the change that is likely to occur within a certain period of time, even if it is not the total duration of function.

It is unthinkable to deny the effect of factors related to prosthetic restoration on the development of peri-implantitis. Factors such as whether the implant carries a crown/bridge/removable restoration, the type of abutment-implant connection or whether the restoration is screwed/cemented may reveal different characteristics on the development of peri-implantitis. According to *Kesar et al. (2023)* the type of prosthetic restoration has been

defined as an independent risk factor for the development of peri-implant diseases. After analyzing the data collected from 274 implants (106 patients) evaluated with an average follow-up of 18 years, they reported that implants carrying removable prosthesis were more prone to peri-implantitis, while implants carrying bridge-type restorations were more prone to peri-implant mucositis. In a study whose results were recently announced, 173 prosthetically loaded implants in 54 patients were followed up for 3, 6 and 12 months (*Nícoli et al., 2024*). This cohort study showed that there was no relationship between the development of peri-implant mucositis or peri-implantitis and the location of the implants, and again, the restorative characteristics of the implant did not affect peri-implantitis. However, due to the insufficient follow-up period and the small sample size and unequal distribution of the subgroups, it provides limited evidence for the evaluation of post-restorative bone loss. Indeed, the authors concluded that there was no relationship between peri-implantitis and the presence of periodontal disease or smoking, which are generally accepted as risk factors. In our study, it was concluded that implants used as support for crown restorations were safer than implants supporting bridge restorations in terms of the development of peri-implantitis. There are several possible reasons for this. First of all, the presence of adjacent natural teeth may also mean that the connective tissue fibers attached to the adjacent teeth are still present in the region and provide some physiological protection. Oral hygiene practices to be performed to remove plaque in crown restorations may be less challenging for patients (depending on the location of the restoration). In addition, the implant (and peri-implant bone tissue), which has to withstand the occlusal force on a single tooth when used as a crown support, will have to withstand a relatively higher load when used as a bridge restoration support and therefore will be more prone to the development of bone loss.

Variation of bone quality and cortical layer thickness in different regions of the mouth are among the main factors affecting the stress distribution in osseointegrated dental implants. The location of the implant is an element that must be evaluated together with different factors such as bone tissue quality, changes in the gingival structure, the magnitude of the occlusal force and the susceptibility to plaque accumulation. The relationship between implant location and the prevalence of peri-implant diseases has been evaluated previously. While some results indicate that there is no relationship, many more studies have shown a significant relationship between prevalence and location. However, there is no clear evidence as to which specific region (mandibular/maxilla or anterior/posterior) has a more effect (*Song et al., 2020*).

This study was based on the collection, analysis and interpretation of data from patients admitted to a university hospital for a year. The fact that the sample consisted of patients admitted to a hospital constitutes a reason for potential selection bias (Berkson's Bias) in the analyses. Because it is not possible to compare the data of individuals who applied to the hospital with those who never applied to the hospital. Additionally, it should be considered that there may be more confounding factors in hospital studies. Ways to avoid this are to not select the sample from the hospital for a retrospective epidemiological study or to design the study as a cohort study from the very beginning. Keeping this in mind, conducting a prospective study that will allow more accurate classification of patients in

terms of demographics, systemic diseases and personal data including habituations will reinforce our knowledge about bone loss around dental implants.

## CONCLUSIONS

Considering the findings of our study, we can conclude that gender, or systemic conditions such as hypertension, hypothyroidism, and diabetes alone do not have an increasing effect on peri-implant bone loss, but smoking can have a significant effect on implant survival. Additionally, advanced age, placing implants in the mandible or using these implants to support bridge restorations can make implants more vulnerable to marginal bone loss.

### Funding
The authors received no funding for this work.

### Competing Interests
The authors declare that they have no competing interests.

### Author Contributions
- Ilkim Karadag conceived and designed the experiments, authored or reviewed drafts of the article, statistical analysis, and approved the final draft.
- Halis Kurnaz conceived and designed the experiments, performed the experiments, prepared figures and/or tables, and approved the final draft.
- Mehmet Murat Akkaya conceived and designed the experiments, authored or reviewed drafts of the article, and approved the final draft.
- İrem Karadag analyzed the data, prepared figures and/or tables, and approved the final draft.
- Zeynep Ilayda Konukçu Kurnaz analyzed the data, prepared figures and/or tables, and approved the final draft.

### Human Ethics
The following information was supplied relating to ethical approvals (*i.e.*, approving body and any reference numbers):

Ankara University Faculty of Dentistry Clinical Research Ethics Committee granted ethical approval to carry out the study (36290600/50).

### Data Availability
The raw data are available in the Supplemental Files.

### Supplemental Information
Supplemental information for this article can be found online at http://dx.doi.org/10.7717/peerj.18643#supplemental-information.

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
