# Peer review of "Observation of peri-implant bone loss rates in patients visiting dentist—A retrospective evaluation of patients of a faculty hospital for one year"

_PeerJ, doi:10.7717/peerj.18643_

## Round 0.1 · original submission · Major Revisions

· Academic Editor

Major Revisions

Your manuscript has been reviewed and assessed by three reviewers, and all of them agree that a few points still need to be addressed. The comments of the reviewers are included at the bottom of this letter. Reviewers indicated that the introduction, methods, and discussion sections should be improved. We would be glad to consider a substantial revision of your work, where the reviewers' comments will be carefully addressed one by one. In addition to these, please see my comments below:

- Line 131: Add “IBM” before “SPSS”.
- What was the power of the study? Which statistical package or tool was used to calculate the sample size? Please provide the name of the package or the tool.
-Line 142: Provide the p-value that was selected for the statistical significance.
-In Table 1-2: Please recalculate the “N”, the total is not equal to %100.
-In Table 3-4: ** shows that p-value is significant or not? Please check the title of the table. If the p-value is significant, please use *; else you don’t need to use that sign.
-If statistical analysis is performed with nonparametric tests, tables and manuscript should include median and IQR (or min-max) values instead of the mean and standard deviation.

Reviewer 1 ·

Basic reporting

I believe that this manuscript is well written and organized, properly referenced.

Experimental design

The study design is well set. Although, I would like to suggest that the authors mention more detailed problematization on systematic desease inluence on dental implants, as well as which systematic deseases are being investigated and why.

Validity of the findings

No comment

Additional comments

No comment

Reviewer 2 ·

Basic reporting

The manuscript is generally well-structured and written in clear, professional English. However, it would be beneficial if the authors could provide more comprehensive background information on peri-implant bone resorption in the introduction section, including more recent literature references.

Experimental design

The study presents original research relevant to the journal's scope, with a clear research question addressing the prevalence of peri-implant bone loss. However, it would be beneficial if the authors could provide more detailed justification for their chosen methodology, particularly regarding the bone loss assessment criteria.

Validity of the findings

The conclusions are generally well-stated and linked to the original research question. However, it would strengthen the study if the authors could provide more robust statistical analysis, particularly for the smoking data where sample sizes differ significantly between groups. Additionally, I would suggest the authors consider discussing the clinical implications of their findings in more depth, especially regarding the observed differences in bone resorption between maxillary and mandibular implants.

Additional comments

(1)It would be better if the authors could acknowledge the potential selection bias inherent in retrospective studies in the discussion section, despite the large sample size (17,352 patients) enhancing the reliability of results.
(2)I would like to suggest further stratifying the patients into different groups based on the duration of implant use (e.g., 3-5 years, 5-10 years, >10 years) to observe if bone resorption changes over time.
(3)It would be beneficial if the authors could explain their rationale for choosing the Froum and Rosen classification method over other common assessment methods, such as millimeter measurements.
(4)I would recommend using weighted analysis or multivariate regression analysis to control for confounding factors in the smoking quantity analysis, considering the differences in sample sizes among groups.
(5)It would be helpful if the authors could provide a more detailed introduction to the etiology and risk factors of peri-implant bone resorption in the introduction section, offering readers a more comprehensive background.
(6)I would like to suggest that the authors mention the limitation of the relatively small sample sizes for diabetes, cardiovascular diseases, and thyroid diseases in the discussion section, and propose larger-scale studies in the future.
(7)It would be valuable if the authors could conduct a more in-depth analysis of the finding that patients smoking 11-20 cigarettes daily had the highest bone resorption scores, despite the difference not being significant, and compare this result with existing literature.
(8)I would recommend exploring potential reasons for the higher bone resorption scores in mandibular implants compared to maxillary implants in the discussion section, such as differences in bone density and occlusal force distribution.
(9)It would be beneficial if the authors could analyze possible causes for the higher bone resorption scores in implants supporting bridge restorations, such as biomechanical factors and cleaning difficulties.
(10)I would suggest discussing the clinical significance of the weak correlation found between age and bone resorption rate in the discussion section, and comparing it with other studies.
(11)It would be better if the authors could mention the limitation of not considering some important factors that may affect bone resorption, such as implant brand, surface treatment, and implantation technique, in the limitations section and include these factors in suggestions for future research.
(12)I would recommend elaborating on the clinical implications of the study results in the discussion section, such as insights for implant placement and restoration design.
(13)It would be valuable if the authors could consider conducting a prospective study combining clinical examinations (such as bleeding on probing and probing depth) with radiographic assessments to more comprehensively evaluate the incidence and severity of peri-implant diseases.
(14)I would suggest analyzing the distribution of different degrees of bone resorption (scores 1, 2, 3), as this could provide more valuable information.

Reviewer 3 ·

Basic reporting

In the manuscript entitled: “Observation of peri-implant bone loss rates in patients visiting dentist – a retrospective evaluation of patients of a faculty hospital for one year" the authors aimed to explore the rates of radiographically detected implants with marginal bone loss and to determine whether there is a relationship between the severity of destruction and patientbased variables.
The authors found that of 17352 patients, 1465 had at least one implant, and 1116 of these had no bone loss. 181 patients (863 implants) included in the study, there was a weak correlation between age and resorption rate. Implants supporting bridge restorations had higher resorption scores. Gender, age, and systemic conditions alone are not eûective in increasing peri-implant bone loss; therefore, placing implants in the mandible or using implants to support bridge restorations may make implants more vulnerable to resorption.

Experimental design

In general, the idea and innovation of this study regards the analysis of factors affecting peri-implant diseases is interesting and novel because the role these aspects in medicine are validated but further studies on this topic could be an innovative issue in this field could be open a creative matter of debate in literature by adding new information.
The study was well conducted by the authors; However, there are some concerns to revise that are described below.
The introduction section resumes the existing knowledge regarding the important factor linked with peri implant diseases, related factors and treatment.

Validity of the findings

The introduction section resumes the existing knowledge regarding the important factor linked with peri implant diseases, related factors and treatment.

Additional comments

However, as the importance of the topic, the reviewer recommends to update the literature through read, discuss some recent interesting articles, that helps the authors to better introduce and discuss the role periodontitis and related mediators and treatment that could worsening peri-implantitis 1) doi: 10.1007/s00784-023-05300-y. PMID: 37814162 2) doi: 10.1186/s12903-023-03237-y. PMID: 37605193
The authors should be better specified, at the end of the introduction section, the rationale of the study and the aim of the study. In the central section, should better clarify inclusions and exclusions criteria of the selected sample.
Please better state the results obtained in the abstract.
The discussion section appears well organized with the relevant paper that support the conclusions, even if the authors should better discuss the relationship regarding by periodontitis treatment approaches that could improve the quality of life in peri implantitis patients. The conclusion should reinforce in light of the discussions.
In conclusion, I am sure that the authors are fine clinicians who achieve very nice results with their adopted protocol.

Minor Comments:

Abstract:
- Better formulate the abstract section by better describing the aim of the study

Introduction:
- Please refer to major comments

Discussion
- Please add a specific sentence that clarifies the results obtained in the first part of the discussion

---

## Round 0.2 · accepted · Accept

· Academic Editor

Accept

Thank you very much for submitting a revised version of your paper. I have gone through the revised, track-changes manuscript and rebuttal letter and see that the authors addressed the reviewers' concerns and substantially improved the content of the manuscript. So, based on my own assessment as an academic editor, the manuscript may now be accepted for publication.

Reviewer 3 ·

Basic reporting

The authors have well-addressed all issues raised by the reviewer

Experimental design

The manuscript can be accepted for publication.

Validity of the findings

-

Additional comments

-